# Development of a Self-Assessment Audit Instrument to Support Climate-Conscious Community Pharmacy Practice and Education

**DOI:** 10.3390/pharmacy11050158

**Published:** 2023-09-29

**Authors:** Stella Fisher, Annalise Mathers, Zubin Austin

**Affiliations:** 1School of Pharmacy, The University of Nottingham, Nottingham NG7 2RD, UK; paysf4@nottingham.ac.uk; 2Leslie Dan Faculty of Pharmacy, University of Toronto, Toronto, ON M5S 3M2, Canada; annalise.mathers@utoronto.ca

**Keywords:** climate change, climate breakdown, climate-conscious pharmacy practice, community pharmacy practice, pharmacy education

## Abstract

Climate breakdown continues to occur at an alarming pace, and the need for all citizens and professionals to respond has never been so urgent. Healthcare work contributes significantly to greenhouse gas (GHG) emissions, representing an opportunity and responsibility for pharmacists to engage in more climate-conscious practices. A key informant interview-based qualitative research method was undertaken to identify options and strategies for community pharmacists to contribute positively to achieving climate-related carbon footprint reduction goals. A total of 10 thought leaders in climate-conscious pharmacy practice were interviewed using a semi-structured protocol, and a constant comparative data analysis method was used to identify common themes and priorities. A self-assessment audit instrument (SAAI) was identified as a positive first step to build awareness, knowledge, skills, and confidence amongst community pharmacists regarding actionable climate-conscious community pharmacy practice. The SAAI supported both self-reflection and self-assessment, while signposting additional resources that could be accessed by pharmacists to continue their learning and professional development. Further work in this area is required if community pharmacy is to contribute positively to climate reduction targets locally and globally.

## 1. Introduction

Climate change and pollution are amongst the most important issues of our time. Climate change is usually defined as the long-term shifts of temperature and other weather patterns on a global scale [1]. Discussion of climate change is most frequently focused on global warming or climate breakdown [2]. For example, the global average increases in temperatures of land and sea surface areas over the past 30 years have had demonstrable impacts on flooding and alterations in wind patterns [3]. Climate scientists have established links between climate change and the burning of fossil fuels, deforestation, and other industrial processes [4]. Since the Industrial Revolution, human activity has resulted in an average increase of 1.5 degrees Celsius in global temperatures—and even higher in polar regions, which are most susceptible to climate breakdown [5].

Pollution is generally defined as the release of toxic or harmful substances into the surrounding environment and encompasses water pollution, soil pollution, and air/air quality pollution [5]. The leaching of chemicals from factories and industrial processes into water bodies, and ecological processes associated with runoff during a precipitation cycle, means that the contamination of the Earth’s fresh water sources is now almost universal and ubiquitous [6]. The World Health Organization estimates that more than 500,000 deaths annually are directly attributable to contaminated fresh water used as drinking water [7]. An even greater concern is air pollution, which is estimated to be responsible for over 7 million premature deaths each year with the highest mortality rates occurring in East Asia and Central Europe [8]. The release of nitrous oxides, sulfur dioxide, and particulate matters pose grave health risks to individuals and planetary health risks on a global scale.

As scientifically trained health professionals, most pharmacists accept the impacts and influence of human activities on climate change and pollution. While pockets of climate skepticism in the profession may exist, the overwhelming consensus of the scientific and clinical communities highlights the realities of climate breakdown and supports commitments to reducing activities that further worsen or contribute to the problem. It is estimated that healthcare work alone contributes close to 5% of all global greenhouse gas emissions and that if healthcare systems were a country, it would be the fifth largest polluter in the world [9]. Despite having a central focus on health, well-being, and healing, the actual work of healthcare professionals like pharmacists contributes meaningfully and massively to the planetary harm that is costing lives, worsening morbidities, and ironically contributing to even more demand for healthcare services that further increase pollution.

Healthcare work contributes to climate breakdown in a variety of different ways. Greenhouse gas emissions (GHGs) occur due to direct emissions from fossil fuel burning and the generation of electricity from non-sustainable/renewable sources. It is estimated that over 70% of the climate footprint of healthcare is a result of GHGs [10]. Within the pharmacy profession, the most common reasons for GHG emissions related to the healthcare supply chain include the production, distribution, storage, and disposal of medicines, devices, or equipment, all of which have significant and sometimes overlooked climate impacts on healthcare [11].

This self-propagating cycle of healthcare work triggering harm was noted in a special 2018 report in the Lancet, which highlighted the realities associated with the significant impact of climate breakdown on global health within the next twenty years [12]. Increased risks of heat-related illnesses, respiratory and inflammatory conditions, contaminated food and water, zoonotic illnesses, and infectious diseases will be even more challenging to manage as extreme weather events such as wildfires, floods, and hurricanes disrupt local and global pharmaceutical supply chains [13,14]. The Canadian Society of Hospital Pharmacists (CSHP) Sustainability in Pharmacy recommendations further highlight the diverse ways in which healthcare work contributes adversely to climate-related health outcomes [15].

The enormity and complexity of this issue means that most pharmacists and pharmacy students may naively assume other large groups or organizations are the only ones capable of enacting impactful change. Some may feel helpless to do anything meaningful to stop this downward spiral, assuming that the actions of individuals will only contribute negligibly to the reduction of carbon footprints. Despite good intentions and a general awareness of the problem, many pharmacists and students may feel that climate breakdown has deteriorated to the point where individuals—or even large employer groups—can have no real impact on stopping or reversing decades worth of planetary harm.

Pharmacy organizations globally have responded to these concerns. For example, the Royal Pharmaceutical Society (RPS) has published its sustainability policies in collaboration with the National Health Service (NHS) in 2021 [16]. They have outlined four main policy areas that are directly relevant to individuals, communities, or hospital pharmacists, including: (i) improving prescribing practices and medicine use; (ii) tackling medicine wastage; (iii) preventing ill health; and (iv) improving infrastructure to reduce carbon footprints and create environmentally sustainable ways of practicing pharmacy. The International Pharmacy Federation (FIP) has also published development goals in 2020: Development Goal #21 focuses on “Sustainability in Pharmacy”, with the goal of transforming pharmacy practice, education, and sciences to align with United Nations Sustainable Development Goals and to contribute positively to the objective of reducing the rate of global heat rise to less than 1.5 degrees Celsius, in line with the Inter-Governmental Panel on Climate Change (IPCC) recommendations [17].

Medicines are the most used intervention in healthcare today; as medication experts, pharmacists have unique opportunities and responsibilities to support planetary health, particularly in jurisdictions (like the UK and Canada) where independent pharmacist prescribing is becoming more prevalent. While macro-level organizational commitments and change from organizations like RPS and FIP is of course essential, providing guidance to individual pharmacists as to how they can also contribute meaningfully to supporting climate change goals is important if the profession is to do its part in reducing the climate impacts of human activity.

The objective of this research was to identify and develop an approach or tool to support individual pharmacists in developing more climate-conscious pharmacy practices. This objective was rooted in the foundational assumption that—as scientifically trained clinicians—pharmacists would not need nor expect to be “convinced” that climate change itself is real, impactful, and problematic. Instead, the research would be framed around the notion that community pharmacists recognize and accept the realities of climate change but are uncertain, unaware, or lack confidence in their own abilities to use their day-to-day clinical practice activities in climate-conscious and impactful ways.

## 2. Materials and Methods

Research Method:

As this research was exploratory in its orientation, a key informant research method was used, and focused on thought, opinion, and practice leaders in climate-conscious pharmacy practice in Canada. This jurisdiction was selected primarily with respect to pharmacists’ scope of practice and abilities to engage in independent prescribing activities. Key informants were identified through a consensus approach, focused on public and professional presence and prominence. For example, members and leaders of grassroots climate-conscious pharmacy practice organizations, or leaders or directors of national pharmacy organizations developing standards or protocols for climate-conscious pharmacy practice, were defined as key informants for the purposes of this study.

The key informant research method has been well established as a tool for exploratory qualitative research such as this, in situations where there is relatively scant literature or information available [18] and where thought leadership can be helpful in identifying opportunities for further exploration [19,20]. In this method, key informants were invited to participate in this research, and upon accepting the invitation, a Zoom-, Teams-, or telephone-based interview was scheduled at a mutually convenient time. A semi-structured interview protocol (Appendix A) was used to guide discussion, but a narrative-analytical questioning style was used to allow key informants to lead discussion in ways that aligned with their expertise rather than slavishly follow a pro-forma script of questions [21].

Verbatim transcripts of interviews were generated and were used to guide the data analysis. A constant-comparative coding method was used to help categorize the data and to establish themes that could guide the subsequent analysis. Data were managed and analyzed using NVivo (v.12). Codes were based on the interview guide questions, then separated into further subthemes, some of which emerged as coding continued. Second checking and the confirmation of coding occurred using a manual data analysis using the same inductive codes. Similarities and differences in coding were discussed until consensus amongst both coders was reached. Interviewing proceeded until the point of thematic saturation, where no new/additional/different concepts or ideas were surfaced.

As thought leaders and prominent pharmacists in Canada, participants in this study volunteered their time to participate; no honorarium or compensation was provided. All participants provided informed consent pursuant to a research ethics protocol approved by the University of Toronto Research Ethics Board. As is customary in key informant interview studies, no demographic information or details regarding participants will be provided in this report as doing so is likely to breach their confidentiality [19,20]; given these individuals’ prominence in the profession, even scant demographic or other information would likely reveal their identities.

## 3. Results

A total of 19 key informants were invited to participate in interviews for this study. As respondents agreed to participate, interviews were scheduled and undertaken at their convenience. A total of 10 interviews were completed prior to thematic saturation being declared. Key informants included executives and leaders with national procurement organizations and large pharmacy employers, academics/researchers, as well as pharmacy managers and pharmacists who had previously presented at national meetings or written in national publications regarding climate-conscious pharmacy practice. Interviews took place in early 2023.

Key consistent themes that were identified in this research included:(a)*The need to enhance general awareness of climate-conscious pharmacy practice amongst community pharmacists:* While key informants agreed that most pharmacists were—conceptually—supportive of climate and heat reduction goals, and aware of climate change as a real and pressing global problem, they noted that most community pharmacists likely felt overwhelmed, helpless, or hopeless about their own potential role in climate-related mitigation or adaptation strategies within a community pharmacy context. While pharmacists did not need convincing that climate change was real and damaging, they needed empowerment and motivational support to believe they could—and should—engage in climate-conscious pharmacy practice activities that would still make meaningful albeit small contributions to achieving climate goals. In part, this may be reflective of many community pharmacists’ own professional identity as a “helper”, a “supporter”, or a “responder” rather than as a “leader”, a “decision maker” or an “initiator” of change. Key informants shared their own trajectories and professional development in this area, noting that as “average” pharmacists, they too had limited awareness of what individual community pharmacists could do, but because of their own passion and interest in climate change and environmental sustainability, they embarked on their own personal learning journeys to fill in knowledge gaps. Key informants also highlighted the importance and value of building communities of practice, and the significant value of grassroots community pharmacy organizations where groups of interested pharmacists could meet and learn from one another, rather than simply await hierarchical top-down lectures from experts who were not actually working as community or hospital pharmacists. Key informants refined and endorsed the notion that a “self-assessment audit instrument” approach to enhancing awareness was appropriate and potentially impactful, noting the value of “carrots” (positive, motivational, or educational approaches) rather than “sticks” (punishments, penalties, or guilt-driven techniques) in building awareness and making a commitment to change. A self-assessment audit instrument that could be used privately by an individual pharmacist to help them reflect upon their current knowledge and skills with respect to climate-conscious pharmacy practice, and that provided abundant links to existing resources and literature to help expand learning and encourage follow-up reading, was conceptualized by key informants as a potentially valuable initial strategy to support pharmacists.

Below are some illustrative quotes from key informants regarding the need to enhance general awareness of climate-conscious pharmacy practice:

“*I think there’s a general lack of awareness of how detrimental our healthcare systems and pharmacies are to the environment, and so I think raising that awareness and then getting pharmacist to actually take action in ways that are feasible for their pharmacy environment*.”

“*The awareness of sustainability and its importance in my practice, and then once you become aware, you start to think about what I am really doing as a sustainable practice*.”

(b)*Waste disposal:* Key informants identified climate-conscious waste-disposal practices as integral to efforts to reduce climate impacts and footprints, and as a specific and unique set of practices for community pharmacists to lead. As discussed previously, the waste-linked pollution of soil, air, and water has devastating public health and climate-related outcomes. Pharmacy has historically been a leader in healthcare with respect to waste management through activities ranging from medication take-back programmes, expanded recycling initiatives, and safe disposal (including sharps disposal) activities. Beyond these traditional activities, however, key informants identified other areas where community pharmacists could exert influence and use their scope of practice to manage waste more effectively. First and foremost were consciousness raising activities associated with the operations of a typical community pharmacy: for example, switching to lower carbon-impact LED lighting systems that provide sufficient illumination with less electricity consumption is an important operational change within the reach of many community pharmacies that reduces electricity waste and consequently reduces fossil fuel consumption. Similarly, timed lighting or motion-detector lighting systems can significantly reduce unnecessary and wasteful electricity consumption. An important and intriguing idea raised by some key informants was related to the notion of “hygiene theatre”, which refers to the array of practices that pharmacists take for granted that are done in the name of sterility, cleanliness, and tidiness. For example, the routine use of alcohol swabbing prior to immunization has been increasingly questioned: while it is taken for granted by patients, pharmacists, and other healthcare professionals that alcohol swabbing reduces risks of infection or other harms prior to injection, there is limited scientific evidence and no clear consensus as to whether this is actually true, particularly for intramuscular injections that many pharmacists routinely administer. If one considers the recent COVID-19 mass vaccination campaign and the hundreds of millions of jabs administered by pharmacists world-wide—and then considers that almost every one of those jabs was first accompanied by the opening of a small non-recyclable foil package, the unfurling of an alcohol swab, the gratuitous swiping of that swab on the skin, and then the disposal of both the foil package and the swab—the environmental footprint of the COVID-19 vaccination campaign becomes unfathomable. Does every person need and benefit from alcohol swabbing prior to vaccination? Is there sufficient microbiological evidence to support such an environmentally impactful routine practice for everyone? Or is this an example of wasteful hygiene theatre designed to provide the illusion of sterility and cleanliness [21,22,23,24,25,26]? Clearly, a further evaluation of this practice is required, but this example of a potentially wasteful practice that is simply assumed to be necessary and beneficial highlights the need to consider more climate-conscious pharmacy practices. Ultimately, key informants noted that much of the waste generation in community pharmacy arises from a lack of thoughtfulness or unreflective practice. Like the alcohol swab example, routine unquestioned practices that generate waste need to be reflected upon more closely to determine alternative options that are less environmentally damaging; raising an awareness of the wastefulness in pharmacy practice is an important first step. A self-assessment audit instrument to highlight these issues, identify opportunities, signpost further resources, and trigger more self-reflection and thoughtfulness around routine practices could have significant benefits in identifying opportunities to reduce, reuse, and recycle in community pharmacy.

Below are some illustrative quotes from key informants regarding waste management:

“*Waste management, how the medications, if they were disposed of properly versus if we don’t really encourage patients to bring back the medications that they don’t need, what kind of environmental impact it could have*.”

“*Implementing a really strong medications take back programme” and “using paper bags instead of plastic”, and “recycling practices overall*.”

(c)*Educating patients*: As the most accessible and trusted of health professionals, community pharmacists have unique opportunities to build sustainable positive relationships with patients in their communities. These relationships form the foundation of the third common theme identified in this research associated with climate-conscious pharmacy practice: focused education for patients highlighting the climate impacts of medications as a way of further broadening knowledge across the population. Pharmacy has already demonstrated some success in initiatives such as this; for example, the Choose Wisely public education campaign highlighted the importance of not taking antibiotics for viral infections by identifying the costs and societal consequences of unnecessary antibiotic use. Initiating similar education campaigns that address issues of the inappropriate, unnecessary, or wasteful use of medications as an environmental issue (not simply a personal health issue) was identified by participants as an important specific task for community pharmacists in supporting climate and heat reduction goals. The pharmacist’s unique ability to engage with and build long term relationships with patients provides unique opportunities to introduce these concepts into the broader community in terms of both personal health benefits as well as planetary health outcomes.

Below are some illustrative quotes from key informants regarding educating patients:

“*Pharmaceutical care plan, being able to tell your patient if he or she is interested and what the carbon footprint is over the next 10 years of you taking this drug because I really think that the Holy Grail and all this is needing less health care………educating people on the potential carbon footprint of whatever medication they are taking*.”

“*So, educating consumers about how to not flush these things down the toilet have not to put them in the landfills, but also their own commercial waste*.”

(d)*Changes in prescribing and dispensing:* Some of the most exciting potential areas where pharmacists could have meaningful impacts on planetary health are connected to the rapidly evolving scope of practice of pharmacists in Canada as well as other countries. As independent prescribing, modification/adaptation of prescriptions, independent therapeutic substitutions, and deprescribing become more commonplace in community pharmacy, there are tremendous opportunities for community pharmacists to leverage their new scope of practice in ways that also benefit planetary health. Even in jurisdictions where pharmacists’ scope of practice does not permit independent prescribing, pharmacists have always used their expertise to assess the “appropriateness” of medications for a patient. Generally, “appropriateness” has been defined in terms of a medication’s efficacy, cost, side-effect profile, potential for drug-drug interactions, etc. Key informants highlighted how the “carbon footprint” of medications could be added to the list of criteria that define appropriateness and pharmacists could incorporate this element into their clinical decision making (or, if allowed, into their prescribing). Specific areas to consider include the kind of packaging used for a medication (recyclable paper vs. non-recyclable foil), and the manufacturing source (e.g., procuring from local manufacturers to reduce carbon impacts associated with international supply chains). The most frequently given example related to therapeutic substitutions is the move away from polluting multi-dose inhalers (MDI) used for asthma or COPD to more climate friendly dry powder inhalers (DPI). The environmental implications of the mass therapeutic substitution of DPI for MDI prescriptions would be significant, and if patients are equally able to tolerate either formulation, pharmacists exercising their scope of practice to independently adapt prescriptions or therapeutically substitute these medications would have considerable carbon savings [26]. Key informants in this study highlighted limitations to this approach: currently there are very limited sources of reputable information regarding the carbon footprints of medications, and few reliable sources that help pharmacists compare the carbon footprints of therapeutic alternatives. This was identified as a critical research and practice support need. Beyond the incorporation of carbon footprints into therapeutic decision making, the growing interest in deprescribing also was identified as an environmental benefit. While deprescribing is primarily driven by the patient’s unique health needs, the positive environmental benefits of reducing the volume of medications consumed is an important additional benefit. Overall, key informants highlighted the significant potential of leveraging pharmacists’ drug expertise to reduce unnecessary polypharmacy for the benefit of both individual patient’s and the environment.

Below are some illustrative quotes from key informants regarding changes in prescribing and dispensing:

“*As well as selecting products that are more environmentally friendly overall, such as inhalers—if a patient is a good candidate for a DPI in place of an MDI, working with the physician and the patient to make that change*.”

“*Other interesting concept is sustainable prescribing—lots of potential here for pharmacists*.”

## 4. Discussion

The key informant interviews provided rich data to support the notion that individual community pharmacists have an important and potentially impactful role to play in combatting climate change, supporting the attainment of the UN and IPCC heat reduction targets, and in mitigating climate breakdown. Based on these interviews, a self-assessment audit instrument (SAAI) for community pharmacists was created. SAAIs are tools created to encourage self-reflection and provide underlying motivation, foundational knowledge, basic skills, and initial self-confidence in undertaking change [21,22,23]. They are traditionally framed as educational support tools: in responding to questions or items on an SAAI, users may be able to compare their results to other similar users for peer benchmarking purposes, and then can be signposted to additional readings or supports to enhance their motivation, knowledge, skills, or self-confidence (depending upon how they responded to the item/question itself). Examples of the use of SAAIs in pharmacy include the Quality Payment Program created by the Centres for Medicare and Medicate Services in the United States [24], and the Medicines Optimisation Assessment Tool (MOAT) developed in the UK [25].

The use of an SAAI in the context of climate-conscious pharmacy practice is unique but aligned with previous applications of this approach. In emphasizing learning, reflection, and self-assessment in a non-punitive and non-judgemental manner, a safe space for learning is provided, as well as a structure for individual pharmacists that allows them to pursue further reading or behavioural change at their own pace, in their own time, and in a manner that is comfortable [22]. As such, SAAIs may be criticized as being too slow, too ineffectual, and having insufficient rigour to promote true change during a climate crisis. Importantly, a more punitive or judgemental approach to awareness building would likely be unsuccessful, resented, or simply impossible to implement, and so the SAAI was selected as an important initial step in creating resources in the years ahead to support climate-conscious pharmacy practice.

Sample items from the SAAI are displayed in Appendix B. As can be seen, the general structure of each item is standardized, involving self-reflection of a practice-related behaviour, self-assessment of frequency, and then an opportunity for the user to learn more about the specific topic explored in this item through links to reputable resources. No peer benchmarking was incorporated at this time due to logistical complexities and a desire to ensure a truely non-judgemental learning and self-reflective environment for the user. The specific content for each item was generated through the key informant interviews and represents the consensus of Canadian thought leaders in the climate-conscious pharmacy practice area. While some items reflect uniquely Canadian contextual points (e.g., the relatively expansive scope of practice of Canadian pharmacists compared to pharmacists in some other countries), much of the content is applicable to community pharmacists anywhere, and most of the signposted resources and readings are international in their orientation.

An initial pilot testing and validation of this instrument have been undertaken and both its value and utility to community pharmacists have been demonstrated. Those involved in the pilot testing indicate that the instrument is easy to understand and use, is effectively non-judgmental, encourages onward reading and professional development, and is presented in an attractive and readable format that encourages pharmacists to use it. Further validation is currently underway to determine what if any outcomes or impacts emerge after pharmacists use the SAAI, and whether sustained behaviour change results. One potential area for further exploration recommended by some key informants was related to the engagement of regulatory/licensing bodies for pharmacy across Canada. The use of this self-assessment audit instrument as a required part of the annual renewal of the registration process could heighten awareness and interest in climate-conscious practice in ways that voluntary continuing education programming may not and reach many more pharmacists than any conference or meeting possibly could. This idea warrants further exploration to determine its value and feasibility.

The role and value of continuing education and continuous professional development to support climate-conscious practice requires further exploration [26,27,28,29]. As awareness of the climate impacts of healthcare work become better known, grassroots organizations and communities of practice in pharmacy are emerging that also provide important potential ways of disseminating this SAAI and other important programming related to climate-conscious practice [30,31]. Collaboration with and the engagement of such organizations and communities will be an essential component of any meaningful interventions [31].

While this study represents an application of the key informant research methodology to the field of climate change, there are limitations to this work. The selection of key informants is inherently subjective and premised on public prominence and professional profile; however, as the objective was to identify influential thought leaders, there are few other viable alternatives. While thematic saturation occurred after 10 interviews, the number of participants in this research was relatively low; however, given the strong degree of alignment amongst thought leaders and amongst these participants, it is not unexpected that saturation would occur after a relatively small number of interviews. Further, these interviews focused on depth and allowed abundant time for deeper exploration and enquiry, generating confidence in the quality of this data. Ultimately, this exploratory study is not the last or definitive word on the topic of climate-conscious pharmacy practice; instead, it is a first step in addressing a complex issue using research as a tool to generate practice-focused interventions to shift pharmacists’ thinking and behaviours. Further work in this area is required and can build on the foundation established in this study.

## 5. Conclusions

The development of a self-assessment audit instrument for climate-conscious pharmacy practice highlights one approach to supporting the pharmacy community in meeting climate goals established by professional associations, governments, and the United Nations. The use of a key informant approach designed around influential thought leaders, and a design approach emphasizing “carrots” (e.g., education, motivation, and awareness building) rather than “sticks” (e.g., judgement, punishment, and sanctions) provides a valuable first step in highlighting the importance of individual actions to address climate breakdowns, not simply relying on large organizations or other people to help address this most pressing issue of our time.

## Data Availability

Due to the nature of this key informant study, data will not be provided as it may compromise the identity of the participants. The unique expertise of the key informants was the reason they were asked to participate in this study; in presenting their comments, it would be impossible to safeguard their anonymity as their unique expertise would disclose their identity.

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
