# Peer review of "Development of a Self-Assessment Audit Instrument to Support Climate-Conscious Community Pharmacy Practice and Education"

_pharmacy, 2023, doi:10.3390/pharmacy11050158_

Round 1

Reviewer 1 Report

Dear authors your work is interesting but it is just a pilot one in the area. You state that your aim is to  identify and develop a model, approach, or tool to support individual pharmacists in developing more climate conscious pharmacy practices. You propose a tools but not the model. Can you please explain.

You interviewed leaders in climate conscious pharmacy practice in Canada. What you mean as leaders? If these people have an academic background or are active in the virtual media. How you define their leadership.

In the results you focused on 4 areas as general perception, wastage, educating patients, prescribing and dispensing. I agree that those are area are very importantr but one of the most important is the regulatory area. In this sence I would recommend to comments on the encouraging or discouraging regulatory measures. 

Author Response

Thank you for taking the time to read and comment on our manuscript.  Your suggestions were reasonable and  helpful.  Specific changes to  manuscript based on your  comments are highlighted below:

  1. We agree that we were too  broad in our aim and indeed did not develop a "model".  We have changed the aim statement to  simply state the objective was to develop a "tool".  Thank you for noting this.
  2. You asked how "leaders" were defined for inclusion in the study. We have added  clarifying text to provide examples based on role (e.g. executives involved in national  procurement  organizations or pharmaceutical manufacturing) or thought leadership in climate conscious practice. (e.g. publishing in national journals or presenting at national conferences).
  3. You suggested we signpost the potential  role of regulator/licensing bodies in supporting this work in the discussion section.  This is a very helpful suggestion and additional  text  has been incorporated to make this point.

We hope these changes are satisfactory and address your concerns.  We believe your suggestions  have improved  the quality of  the manuscript and we thank you for providing them.

Reviewer 2 Report

The authors developed a qualitative study with the aim of developing a model, approach, or tool to support individual pharmacists in developing more climate-conscious pharmacy practices. However, the results only show the results of a qualitative study with key informants and do not present the model developed ... only three sample items from the SAAI are displayed in Appendix 2. Also, the methods section does not describe how the model was constructed, all the framework should be described.

The introduction is too long. The authors should focus more on justifying the importance of pharmacists' awareness of environmental issues, in order to justify the need to develop the model. It is important to add bibliographical references in the introduction because some descriptions are not supported by a bibliography. some examples are Lines 49-59 and Lines 69-754.

The objective of the work I presented in the methods section, but usually it is presented in the final of the introduction. I suggest moving it.

It is not clear if the themes emerged from the interviews or were predefined. During the results section authors present a description for each of them that is not well supported by the quotes presented. It is necessary to add more quotas that support the discussion.

For more clarification, I also suggest a table with the themes and respective quotes. 

Author Response

Thank you for reviewing and commenting on our manuscript.  We appreciate both the time and care taken to do this, and the quality of your comments which helped improve the manuscript.  Specific responses to your suggestions are noted below:

  1. You note that in our objectives we set out to  develop a "model,  approach or tool", when in fact all  we  do  is present a tool.  We agree with your comment that the objective is too ambitious and broad, and we have revised the objective  statement accordingly to  indicate we are focused on developing a tool.
  2. You note the introduction section is too long.  We have removed the equivalent  of approximately two full  paragraphs of content from the introduction section which we hope will enhance readability without compromising important background information.  Thank you for this important recommendation.
  3. You suggested changing the place where the objective for this project was  presented, to the end of the introduction section.  We have done this.
  4. You suggested adding more quotes to support the discussion; while we agree this would be helpful,  we  are concerned that the manuscript is already quite long and additional quotes or clarifying data would lengthen the manuscript even further so have respectfully chosen to not act on this suggestion, despite our acceptance of the  point you are making.

We hope these clarifications  are satisfactory to you.  Your comments have been very helpful and have improved the quality of our work - thank you.

Reviewer 3 Report

I would like to say, that this is a very good study delving in to the topic that is just beginning to be discussed with more emphasis. The tool developed by authors can be useful for self-reflection purposes in pharmacy practice. I would like to raise a few points that could improve the manuscript. 

1) The introduction provides a good background to topics reported by interviewees. It could be shortened a bit, in order to improve readability (part in lines 75-85). 

2) Authors performed their study in canada. Are there some local guidelines or statements, that could be cited reffering to the topic of the manuscript?

3) Authors do not provide information on informants in order to safeguard anonimity. Still, some data could be provided without breaching confidentiality, data that regards the group, not individuals, e.g. median age of professional practice, number of pharmacy managers vs. pharmacists. 

4) Authors did not provide data they gathered on skills, attitudes, and behaviours in respects to environmentally sustainable practice reported by informants (Question 4 in Appendix 1). I think it could be important to describe these results as the research may contribute to development of educational materials on sustainable pharmacy practice. 

5) Issue of inhalers: wrong abbreviation is given in line 321. Authors could also reffer to literature discussing the issue, e.g https://doi.org/10.1002/prp2.675

6) Issue of waste disposal. Authors should reffer to respective policies in Canada, are they sufficient? Also, the issue of alcohol swab should be reffered to giving some literature example, e.g. doi: 10.1016/j.vaccine.2020.12.082

7) In the Discussion, authors could reffer to other means to raise awereness on the problem of sustainable pharmacy practice, like undergraduate and postgraduate or CPD courses. Please reffer readers to some good literature on the topic. One of the chosen keywords is 'pharmacy education' and I think the reaedrs could appreciate it. 

8) Communities of practice. Informants reported that this could be a good idea to use in pharmacist education. I think that this is valuable suggestion and the authors should discuss the issue in Discussion. How could comminities of practice be implemented, are there some examples that could be followed?

Author Response

Thank you for the detailed and helpful comments and recommendations.  Your suggestions were very insightful and we are in agreement with all of them; below, please find more detailed descriptions of how we implemented your recommendations in the revised manuscript.

  1. You suggested reducing the length of the introduction section.  We agree  and have now removed the equivalent of two full paragraphs of content to reduce the density of this section without interfering with the important content.
  2. You noted that since the study context was Canada, that additional  guidelines, statement, or commentary from Canada should be included - we have added three additional references from various Canadian sources (the Canadian Society of Hospital Pharmacists, the Canadian Pharmacy Journal and the Canadian Agency for Drugs and Technologies Assessments) to address this concern.
  3. You suggested some clarifying text be added to indicate who are key informants were and why they were selected, in ways that would not compromise their anonymity.  We have added additional text indicating informants were primarily selected based on their role (e.g. executives/leaders in national procurement agencies or pharmaceutical manufacturers) or their prominence as thought leaders (e.g. through publication in national journals or presentations at national conferences).
  4. You noted that we did not provide data related to Question 4 of the semi-structured interview guide focused on skills, attitudes or behaviours.  This was a difficult decision but primarily related to the additional length that this would have added to an already long manuscript.  The data from question  4 may eventually be used in a more educationally-focused paper but was not included in this manuscript primarily for length reasons.
  5. Thank you for  noting the incorrect abbreviation at line 321 and the suggestion for a reference to support the claim - these changes have now been incorporated in the manuscript.
  6. Thank you for your comments regarding referencing as it relates to waste disposal.  We have now incorporated Canadian references as well as a specific Canadian reference with respect to the alcohol swab issue.  
  7. You note the value of  additional references related to CPD and other courses,  as well as the potential of communities of practice.  Additional clarifying text and references have now been added to the paper.

Thank you for your helpful suggestions and thoughtful comments.  They have helped to improve the quality of our manuscript.

Reviewer 4 Report

Few pharmacists have an awareness and working knowledge of climate conscious pharmacy practice despite the critical need. This manuscript describes the creation of a self-assessment instrument that can be used by practicing pharmacists aimed at increasing awareness of climate conscious pharmacy practice. The instrument was created using a key-informant qualitative methodology, which was clearly described in the paper. Key informants were selected from Canada based on their presence in the public and the profession. Through semi-structured interviews, this work identified four major themes that informed the creation of the self-assessment instrument: the need to enhance awareness of climate conscious pharmacy practice, the need for proper waste disposal practices, the necessity of patient education, and the impact of changes in prescribing practices. This very well-written paper describes the development of this novel self-assessment audit instrument for pharmacists that will enhance awareness and conversations regarding climate conscious pharmacy practice around the globe. 

I found this paper to be clearly written, informative, and comprehensive. The methodology was well-described and scientifically sound. Given the current state of global climate change, this paper is incredibly timely and should be of interest to all pharmacists and healthcare leaders. I have but a few minor comments aimed at strengthening the manuscript prior to publication.

1. Line 37. The words, "In contrast..." are probably not necessary, as it is evident that climate change and pollution are two different but related concepts. I would probably just start a new paragraph here.

2. Line 59. I would suggest changing the end of this statement to read, "... demand for healthcare services that further increase pollution."

3. Lines 75-85. The introduction is a bit long and becomes rather redundant in this paragraph. I would suggest taking any new information from this paragraph and reincorporating it into other parts of the manuscript while deleting statements that are redundant.

4.  Lines 100-107. I feel like the flow of the introduction might improve if this paragraph was moved to follow the paragraph in lines 108-120.

5. Line 441. Would it be better to include this as a figure instead of an appendix?

Author Response

Thank you for taking the time to  review and comment on our manuscript.  Your supportive and constructive feedback is appreciated and we are gratified by your positive  response.  Specific changes made to the manuscript based on your feedback are noted below:

  1. Line 37 - as you suggested, we have removed "in contrast",  and started a new paragraph where discussion of pollution is introduced.
  2. Your suggested rewording for Line 59 was very helpful, and has been accepted - manuscript amended accordingly.
  3. Your comments regarding the length and flow of the introduction were helpful.  We have now removed the equivalent of two full paragraphs from the introduction section as a way of managing redundancy, and have accepted your suggestion to move lines 100-107 to now follow lines 108-120.  Thank you for this very specific and helpful feedback.
  4. Your comment as to whether "Figures" instead of "Appendix" should be used has not been acted upon yet - we will await final confirmation of the preference of the journal with respect to naming.

Thank you again for your very helpful comments - they have improved the  quality of our manuscript.